# Impact of the Nucleic Acid Extraction Method and the RT-qPCR Assay on SARS-CoV-2 Detection in Low-Viral Samples

**DOI:** 10.3390/diagnostics11122247

**Published:** 2021-11-30

**Authors:** Magdalena Komiazyk, Jarosław Walory, Aleksandra Kozinska, Izabela Wasko, Anna Baraniak

**Affiliations:** Department of Drug Biotechnology and Bioinformatics, National Medicines Institute, 00-725 Warsaw, Poland; j.walory@nil.gov.pl (J.W.); a.kozinska@nil.gov.pl (A.K.); i.wasko@nil.gov.pl (I.W.)

**Keywords:** COVID-19, SARS-CoV-2, diagnostic, RNA extraction method, RT-qPCR assay

## Abstract

COVID-19 was initially reported in China at the end of 2019 and soon thereafter, in March 2020, the WHO declared it a pandemic. Until October 2021, over 240 million COVID-19 cases were recorded, with 4.9 mln deaths. In order to stop the spread of this disease, it is crucial to monitor and detect any infected person. The etiologic agent of COVID-19 is a novel coronavirus called SARS-CoV-2. The gold standard for the detection of the virus is the RT-qPCR method. This study evaluated two RNA extraction methods and four commercial RT-qPCR assays routinely used in diagnostic laboratories for detecting SARS-CoV-2 in human specimens from the upper respiratory tract. We analyzed a panel of 70 clinical samples with varying RNA loads. Our study demonstrated the significant impact of the diagnostic methods selected by the laboratory on the SARS-CoV-2 detection in clinical specimens with low viral loads.

## 1. Introduction

Coronavirus disease 2019 (COVID-19) was initially reported in Wuhan, China, in December 2019 and spread to other countries within a few months. On March 11th, the outbreak was declared a global pandemic [1]. Until October 2021, over 240 million COVID-19 cases were recorded, with 4.9 mln deaths [2]. The etiologic factor of this infectious disease is a coronavirus classified by the World Health Organization (WHO) as Severe Acute Respiratory Syndrome Corona Virus 2 (SARS-CoV-2) [3]. There are currently seven coronaviruses (CoVs) that cause human respiratory diseases, among which SARS-CoV, MERS, and SARS-CoV-2 have caused significant outbreaks with high mortality [4]. Genome sequencing analysis showed that SARS-CoV-2 shares 88% identity to two bat SARS-like CoVs (bat-SL-CoVZC45 and bat-SL-CoVZXC21), 79% identity to SARS-CoV, and 50% identity to MERS-CoV [5].

COVID-19 can range from asymptomatic to severe or even critical infection with mortality [6,7,8,9]. The most common COVID-19 clinical symptoms are fever, cough, fatigue, chest pain, headache, and shortness of breath; less frequent are anosmia and diarrhea. A lot of them are characteristic of numerous other viral diseases, including influenza and other respiratory tract infections. Therefore, molecular diagnostics is essential to detect the virus in human samples.

The quantitative (real-time) reverse transcription-polymerase chain reaction method (RT-qPCR) has been introduced as a gold standard in SARS-CoV-2 diagnostic. There are many commercial RT-qPCR kits for detecting SARS-CoV-2 approved by the WHO and commonly used in diagnostic laboratories worldwide [10]. The easy availability and high sensitivity of this test allowed the detection of even a low amount of SARS-CoV-2 RNA in a sample. In the majority, primers are designed to detect RNA fragments from five SARS-CoV-2 RNA regions, including envelope (E-gene), spike (S-gene), nucleocapsids (N-gene), RNA polymerase-dependent RNA (RdRp) and Orf1ab [11]. The commercially available assays differ in their sensitivity and specificity related to applying a range of primers specific to various fragments of the regions mentioned above [12,13]. Therefore, it is important to verify as many commercial and used RT-qPCR tests as possible. It will enable to choice of the appropriate, not necessarily the most popular test, matched to the possibilities of the diagnostic laboratory.

The primary source of errors leading to improper diagnostic of SARS-CoV-2 was described by Rahbari et al. [14] and divided into three groups: pre-analytical, including sampling steps; analytical, including nucleic acid extraction methods and RT-qPCR assays; and post-analytical, focusing on the misinterpretation of the results. Therefore, it is crucial to choose appropriate research methods and interpret the results. The results may also be false-positive or false-negative [15,16,17,18], so proper interpretation by physicians is very important.

Our study demonstrated the impact of the RNA isolation method and the kind of RT-qPCR assay used on the SARS-CoV-2 identification in clinical specimens with a varied load of viral RNA. Two RNA extraction procedures (column- and magnetic-based) and four commercial RT-qPCR tests from different companies were analyzed. Furthermore, we have shown that the lower the viral load, the more difficult it is to interpret the results. The problem with the available diagnostic tests is that they fail to detect the actively infecting virus.

## 2. Materials and Methods

### 2.1. Clinical Samples

Seventy clinical samples collected between May and December 2020 were analyzed. The specimens originated from five hospitals and one drive-thru mobile collection site in six different cities in Mazovia, Poland. All samples were nasopharyngeal cavity swabs transported in different virus-dedicated media. After the diagnostic tests, the samples were stored at −80 °C, pending further studies.

### 2.2. RNA Extraction Procedures

Viral RNA was extracted using two complementary methods. The first method was column-based RNA extraction with a Viral DNA/RNA kit (A&A Biotechnology, Gdansk, Poland). The extraction was performed from 100 µL of samples. The elution volume was 35 µL. The other RNA extraction was performed using a magnetic-based method with a NucleoMag Pathogen kit (Machery-Nagel, Duren, Germany). This kit is suitable for both automatic and manual extraction. In this case, we used the manual method with magnetic blocks. However, results in both automatic and manual methods should be comparable. The volume of clinical samples was 200 µL, and RNA was eluted in 50 µL elution buffer. Each method was performed according to the manufacturer’s recommendations.

### 2.3. RT-qPCR Assays

Detection of SARS-CoV-2 virus was performed by four commercial RT-qPCR tests commonly used in Poland: Bosphore Novel Coronavirus (2019-nCoV) Detection Kit v3 (Anatolia Geneworks, Istanbul, Turkey), DiaPlexQ Novel Coronavirus (2019-nCoV) Detection Kit (SolGent, Daejeon, Korea), Liferiver Novel Coronavirus (2019-nCoV) real-time Multiplex RT-PCR Kit (Liferiver Biotech, Shanghai, China), and MutaPlex Coronavirus real-time-RT-PCR-Kit (Immundiagnostik, Bensheim, Germany). The characteristics of the kits are shown in Table 1. All PCRs were carried out according to the program recommended by the test manufacturers (Table 2) using the Applied Biosystems QuantStudio 6 Pro Real-Time PCR System (Life Technologies Holdings Pte Ltd., Singapore). The results were interpreted based on the quantification cycle (Cq) value according to manufacturers’ recommendations (Table 1).

Two of described RT-qPCR kits (DiaPlex and MutaPlex) along with both RNA extraction methods were subjected to external evaluation (Instand e.V., Dusseldorf, Germany) and passed the positive verification.

### 2.4. RT-qPCR Efficiency

The RT-qPCR efficiency was determined using total human RNA containing RNA of SARS-CoV-2 with an estimated E-gene copy number of 2 × 10^5^ copies/µL (in-house standard of RNA). The number of E-gene copies was determined using the standard curve, obtained by RT-qPCR of the serial dilution of RNA containing a known number of E-gene copies using the MutaPlex Kit. The duplicate 10-fold dilution series of the in-house standard RNA was run using each RT-PCR kit. The PCR efficiency was determined using the Design and Analysis Quant Studio 6 Pro software. The experiment was performed in three technical replicates.

### 2.5. Limit of SARS-CoV-2 RNA Detection

The limit of SARS-CoV-2 RNA detection was determined using RNA extracted from 10^4^ copies/mL of SARS-CoV-2 (Inactivated SARS-CoV-2 Whole Virus Control; Microbiologics, St. Cloud, MN, USA). The RNA was extracted using the column-based method. The range of a two-fold series dilution (copy number 800-8) in two technical replicates was run using each RT-PCR kit. The lowest detected number of RNA copies was determined as the limit of virus detection.

### 2.6. Detection of SARS-CoV-2 in Clinical Samples and Quality of RNA

RNA from 70 clinical samples were extracted using two complementary methods, as described in Section 2.2. This was followed by SARS-CoV-2 identification using four RT-qPCR kits identifying SARS-CoV-2, according to the methods described in Section 2.3. The samples were analyzed under the procedure developed by our medical laboratory. As the specimens were taken once from the patients, only one measurement was done from each sample.

The quality and amount of extracted RNA were estimated based on the Cq value of ISC and IPC measured during the RT-qPCR by MutaPlex kit. The arithmetic mean ± standard deviation of the population was calculated. The mean Cq of RNA extracted by the two tested methods were compared using one-way ANOVA, followed by the Bonferroni–Holm post hoc test [19]. Statistically significant differences between groups were defined as *p*-values less than 0.05.

### 2.7. Statistical Comparison of RNA Extraction Methods and RT-qPCR Tests

Comparisons for paired nominal data were performed using chi-square McNemar’s Marginal Homogeneity Test. A *p*-value < 0.05 was considered statistically significant. Statistical analysis was performed with Statistica software 9.0 (Stat Soft Inc., Tulsa, OK, USA).

### 2.8. Clinical Sensitivity and Specificity

The reliability of the studied diagnostic tests was analyzed using the module *Evaluation of a diagnostic test,* PQStat software (Poznan, Poland) where confidence intervals were calculated based on the Clopper-Pearson method for a single proportion. As true positives, samples that were positive in at least two tests were considered. The total number of samples tested was 70, of which 53 were True Positives (TP) and 18 True Negatives (TN). The inconclusive samples were classified as positives.

### 2.9. Detection of Different SARS-CoV-2 Variants

RNA from various SARS-CoV-2 variants, including B.1.617.1 (one sample), B.1.617.2 (five samples) and B.1.1.7 (seven samples), was isolated using the column-based method and detected by the studied RT-qPCR tests, as described above. The obtained RNA is a part of the National Medicines Institute Collection and has been identified by commercially available SARS-CoV-2 mutation detection assays (data not published). The experiment was performed in three technical replicates.

## 3. Results

### 3.1. Influence of the Extraction Method on the RNA Quality

The RNA was extracted from the same samples using two complementary methods, column- and magnetic-based. Each test has a control of reverse transcription inhibition, and an internal control (IC). Thus the influence of inhibitory factors on the result can be monitored. MutaPlex has additional control of the amount of isolated material. It contained primers and probes for β-actin, sample control (ISC). The amount of β-actin correlates with the amount of the genetic material in the sample. These controls allow monitoring the quality and quantity of the isolated RNA.

The result obtained from the RT-PCR by MutaPlex allowed determining no significant difference in the mean Cq for the amount of isolated material in the entire population of the extracted RNA with both methods. The Cq mean for column-based method was 25.82 +/− 2.8, while from magnetic-based method 26.17 +/− 2.7 (n = 70, *p* = 0.44). However, in a few samples, we observed ten times lower RNA concentration of RNA extracted by the column-based method related to the magnetic-based method, which did not affect the detection of the SARS-CoV-2, Figure 1. In samples 17, 20, 23, 27, 50, 60, 62, the amount of extracted RNA was similar for both methods, but the virus was detected in samples isolated by the column method. On the other hand, in samples 31, 59 and 65, the amount of isolated genetic material was higher in the magnetic-based method, but the virus was not identified. Thus, the amount of extracted RNA did not influence the SARS-CoV-2 detection.

### 3.2. Efficiency of the RT-qPCR Assays

A serial dilution of the SARS-CoV-2 RNA (10-fold dilution over 5 logs) was run with the studied RT-qPCR assays to evaluate their efficiency. For each kit, the R square was > 0.99, with an efficiency above 90% (Figure 2). The efficiencies of Bosphore’s E-gene and Orf1ab and Liferiver’s Orf1ab were slightly higher than 110%, which can be related to reaction inhibition.

### 3.3. SARS-CoV-2 Detection Limits of the RT-qPCR Assays

To determine each kit’s detection limit, the inactivated SARS-CoV-2 whole virus control was used. This control sample contains the whole inactivated SARS-CoV-2 particles, with a viral load of 10^3^ copies/sample. Following the RNA extraction, we got around 250 gene copies/µL (estimated using a standard curve for E-gene detected with MutaPlex).

The DiaPlexQ and Liferiver kits exhibited a similar detection limit. Both detected at least one target gene, in samples with less than 16 viral copies. Although in samples with a low viral load, only single genes were detected but results were not repetitive.

The detection limit for MutaPlex, was 32 gene copies/sample, of which only E-gene was detected in this dilution. In the case of the Bosphore assay, the detection of fewer than 63 copies of E-gene could not be observed (Table 3).

### 3.4. Impact of RNA Extraction Method on the Detection of SARS-CoV-2

Comparison of RNA extraction methods demonstrated that the column-based method was more efficient than the magnetic-based method for three of four assays tested (Figure 3). However, for the Bosphore RT-qPCR kit, the magnetic method was more effective than the column-based (Figure 3a). Fragments of SARS-CoV-2 RNA (both, positive and inconclusive results) were detected in 18 samples in material extracted by columns and 31 samples extracted via magnetic-based methods. This means that 42% more positive samples were detected by magnetic beads extraction than with the second method, thus, both methods differed significantly (*p* = 0.00087). The SARS-CoV-2 RNA extracted by the column-based method was detected, using the DiaPlexQ kit, in 52/70 (Figure 3b), Liferiever in 62/70 (Figure 3c), and MutaPlex in 51/70 samples (Figure 3d). In the case of the magnetic-based method, samples containing SARS-CoV-2 RNA were fewer by 15, 19, and 13, respectively. Based on the obtained results, it was noticed that for each of these three RT-qPCR tests, the column method affects the detection of SARS-CoV-2 in a greater number of positive samples, and this difference is statistically significant (*p* = 0.00132, *p* = 0.00009, *p* = 0.01616, respectively).

The highest differences in the efficiency of RNA extraction methods were observed in samples with low viral load, e.g., less than 500 copies/sample (Cq > 30) (Figure 3). Most samples with high viral load (< 30 Cq) were correctly identified, without significant differences between the applied extraction methods and the RT-qPCR kits.

### 3.5. Reliability and Comparison of the Studied Diagnostic RT-qPCR Kits

One of the goals of this work was to compare the SARS-CoV-2 detection sensitivities of four commercial RT-qPCR kits using a panel of 70 clinical samples. The identification rate depends on the RNA isolation method and the RT-qPCR kit (Figure 3 and Figure 4).

The comparison of the reliability of studied tests is summarized in Table 4 and Table 5. The highest sensitivity was observed for the Liferiver kit, for the column-based extraction method was 98%, while 79% for magnetic-based. A slightly lower sensitivity was observed for a MutaPlex (94% and 75%) and Diaplex kits (96% and 68%). The lowest sensitivity has a Bosphore kit, and it was only 34% for column-based and 58.5% for other RNA extraction methods. The specificity was the highest for the Bosphore kit (100%), while the lowest for Liferiver with RNA extracted by column-based method (41%).

The Bosphore RT-qPCR kit identified 13 positives and five inconclusive samples with RNA extracted by columns, detecting 26% of samples containing SARS-CoV-2 in the specimen set tested. The detection sensitivity in samples with RNA extracted using the magnetic-based method was higher than the column; 18 positives and 13 inconclusive samples were identified (Figure 3a). However, this kit did not detect samples with low viral load, i.e., less than 500 copies/sample (Cq > 30).

The DiaplexQ and MutaPlex kits showed similar effectiveness (Table 5). The MutaPlex kit detected 51 positive samples after RNA extraction by the column-based method and 41 samples using the magnetic method (Figure 3d).

Using the DiaPlexQ kit, SARS-CoV-2 RNA was detected in 50 samples with RNA extracted by columns and in 37 samples with RNA extraction by the magnetic method. Additionally, two samples were inconclusive after RNA extraction with columns, which means that only the N-gene was amplified (Figure 3b).

The highest sensitivity had the Liferiver kit. In probes with RNA extracted by columns, 54 were identified as SARS-CoV-2 RNA-positive. Eight samples were considered inconclusive because amplification was observed only in one gene. In contrast, 42 positives and four inconclusive samples were identified in the material extracted by the magnetic-based method (Figure 3c).

### 3.6. Identification of the Most Common SARS-CoV-2 Variants

The possibility to detect SARS-CoV-2 variants, including B1.1.7, B.1.617.1 and B.1.617.2 by RT-qPCR was verified. Mutations in the above variants did not influence the detection by the used RT-qPCR kits.

## 4. Discussion

This study focuses on applying various RNA extraction methods and RT-qPCR kits to detect SARS-CoV-2. There are three main ways of viral RNA extraction used in diagnostic laboratories worldwide [14]. The first is entirely automatic and requires special equipment, which is not available in each laboratory. The two other are manual, column, and magnetic bead-based methods and do not require advanced techniques. The first method is mainly used in developed countries by routine clinical laboratories. In contrast, smaller laboratories temporarily converted into covid units use manual methods.

Two manual RNA extraction methods were analyzed. The obtained data suggest that SARS-CoV-2 is more frequently identified in the clinical samples with RNA extracted by the column-based method. All samples with more than 10^3^ copies of SARS-CoV-2 RNA were correctly identified in both cases. The main problem appeared when the viral load was low, i.e., less than 500 copies (Cq > 30). In this case, more positive samples were identified when RNA was extracted using columns. The influence of the applied RNA extraction method was also demonstrated by Ambrosi et al. [20]. They tested four different RNA extraction kits and noticed that kits should be validated against the intended use. The conclusion was that maximal caution is needed when detecting SARS-CoV-2 genes at high Ct values, and re-testing should be recommended [20].

It is crucial to remember that samples with low viral load (high Ct values) concern two cases. The first relates to specimens from patients coming out of viremia or who have recently had COVID-19. For these patients, failure to detect viral RNA is no more relevant in relative safety. The problem occurs with the second group of patients, those who are just beginning to enter viremia. In them, failure to detect SARS-CoV-2 RNA will result in these individuals being unwitting carriers of the virus and spreading it. Obtaining a negative test will undoubtedly relieve them of the suspicion of having COVID-19, and they will attribute all potential symptoms to other diseases. However, obtaining an inconclusive test result obliges repeating the test after a specified period (in Poland, it is 24–48 h). After this time, the patient can obtain a conclusive (positive or negative) test result [8].

Four different commercial assays were performed to investigate how the choice of the test influences the identification of specimens with SARS-CoV-2. The three analyzed methods provided comparable results. The RT-qPCR assay from Liferiver showed the highest sensitivity but the lowest specificity. This kit allowed us to identify the most significant number of positive samples, with few inconclusive samples. However, such an increased sensitivity of the test is not an ideal solution in diagnosing the virus. The presence of a few copies of the virus in a sample may be caused by contamination, e.g., during swabbing or sample processing [21]. Inactive viral fragments can also be detected months after the infection, resulting in false-positive results and unnecessary quarantine of healthy patients [18,22]. Thus, it is good practice to repeat the swab and test if a small amount of viral RNA is found in the sample. If the viral load increases after 24 h, active viremia is likely. If it decreases or does not change significantly, an active infection can be excluded. It was reported that patients could not be contagious with Cq > 25 as the virus is not detected in culture above this value [23,24]. Another study demonstrated that at Cq = 25, up to 70% of patients remain positive in culture and that at Ct = 30 this value decreased to 20%. At Cq = 35 less than 3% of cultures are positive, but this should not impact public health decisions [24]. Thus, information about the viral load may be helpful for physicians and epidemiologists. As the Cq values varied significantly among the methods, a more descriptive criterion could be used. To facilitate result interpretation, the method proposed by Carrol and McNamara, reporting the Cq value ranges in ‘high’, ‘medium’, and ‘low’ categories, seems to be promising [25]. However, to apply this criterion, it would be necessary to prepare appropriate global standards based on the viral load. Changing the reporting of results would then be more helpful in diagnosing patients.

The selected diagnostic method is of great importance for the obtained results. Therefore, it is significant to choose a technique that matches the capabilities and equipment in a given laboratory. Many ready-made tests contain information on method validation for a specific RNA extraction method and RT-qPCR equipment. It is possible that the applied RNA extraction method is not compatible with the RT-qPCR kit, as it could be with the RT-qPCR Bosphore kit. Preliminary data with the Bosphore kit showed that it is susceptible to ethanol or protein contamination (unpublished data), commonly inhibiting RT-PCR [14]. Therefore, it seems that selecting an appropriate RNA extraction method is equally essential as choosing the proper RT-qPCR-detecting SARS-CoV-2 kit.

Almost two years have passed since discovering SARS-CoV-2, and it is still a considerable challenge for patients, physicians, and diagnosticians. New mutations appear almost daily, posing significant difficulties in virus detection [26]. To stop the spread of the virus, proper diagnosis and subsequent isolation of the infected person are extremely important. Thus, diagnostic laboratories must follow new reports and carefully observe and control them via RT-qPCR tests.

## Figures and Tables

**Figure 1 diagnostics-11-02247-f001:**
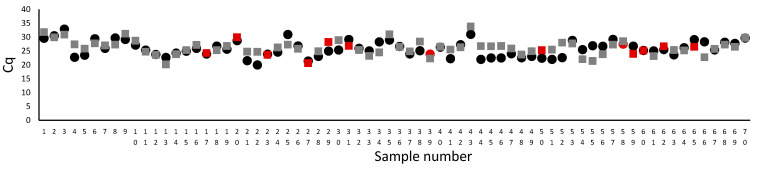
The comparison of Cq for β-actin, which corresponds to the amount of isolated genetic material. The circles represent the magnetic-based, while squares column-based RNA extraction method. The Cq mean for the column-based method was 25.82 +/− 2.8, while from the magnetic-based method 26.17 +/− 2.7 (n = 70, *p* = 0.44, the comparison was made by one-way ANOVA, followed by the Bonferroni-Holm post hoc test). The red marker shows the sample for which the SARS-CoV-2 was detected only in RNA extracted by one method (samples 17, 20, 23, 27, 29, 31, 39, 50, 58, 59, 60, 62, and 65; red indicates positive, while the standard color is negative). Samples were analyzed with MutaPlex kit.

**Figure 2 diagnostics-11-02247-f002:**
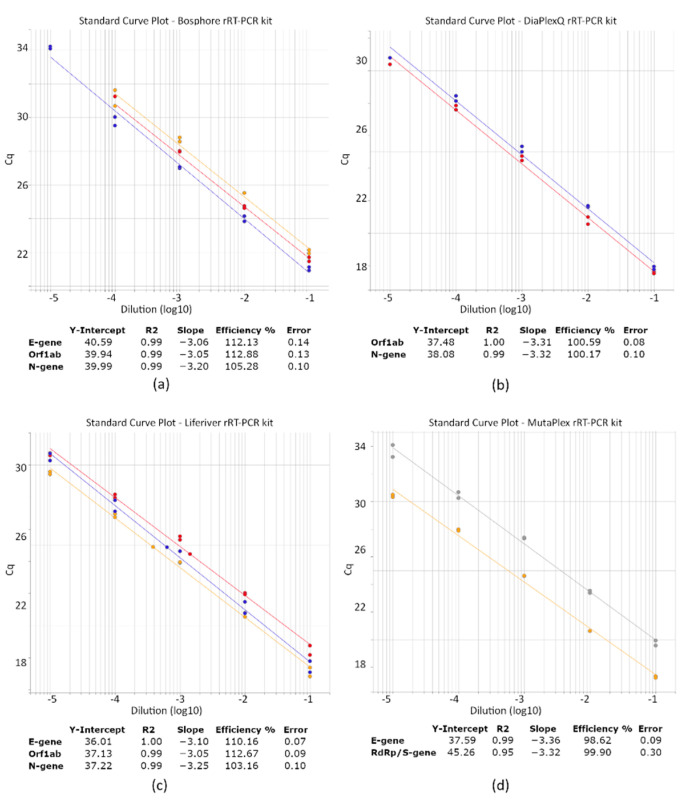
PCR efficiency of the studied RT-qPCR kits detecting SARS-CoV-2. PCR efficiency was analyzed using a duplicate 10-fold dilution series of the standardized clinical sample containing SARS-CoV-2 viral RNA. Linear regression for each target gene: E-gene (yellow line), N-gene (blue line), Orf1ab (red line), and RdRp (grey line) was performed in Quant Studio 6 Pro to obtain the slope and R2. Panel (**a**) refers to the efficiency of the Bosphore, (**b**) DiaPlexQ, (**c**) Liferiver, and (**d**) MutaPlex kit.

**Figure 3 diagnostics-11-02247-f003:**
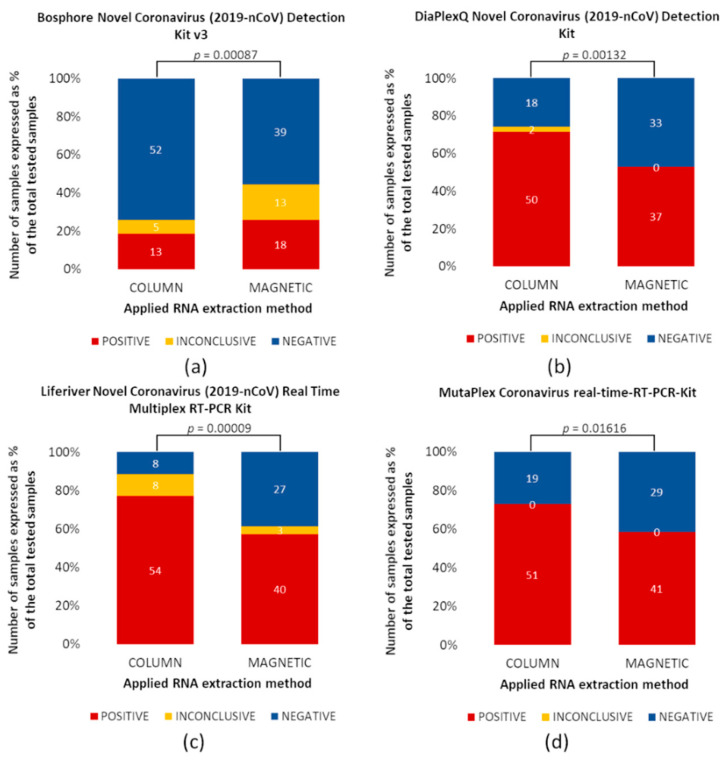
Comparison of two RNA extraction methods. RNA was extracted using two methods, column- (left) and magnetic-based (right). All samples were analyzed using four different RT-PCR kits identified SARS-CoV-2, (**a**) Bosphore, (**b**) DiaPlexQ, (**c**) Liferiver, and (**d**) MutapPex. Graphs present the % of positive (red), inconclusive (orange), and negative (blue) samples with a total of 70 samples. Samples for which SARS-CoV-2 RNA was not detected are marked as 0. Comparisons for paired nominal data were performed using chi-square McNemar’s Marginal Homogeneity Test. A *p*-value < 0.05 is statistically significant.

**Figure 4 diagnostics-11-02247-f004:**
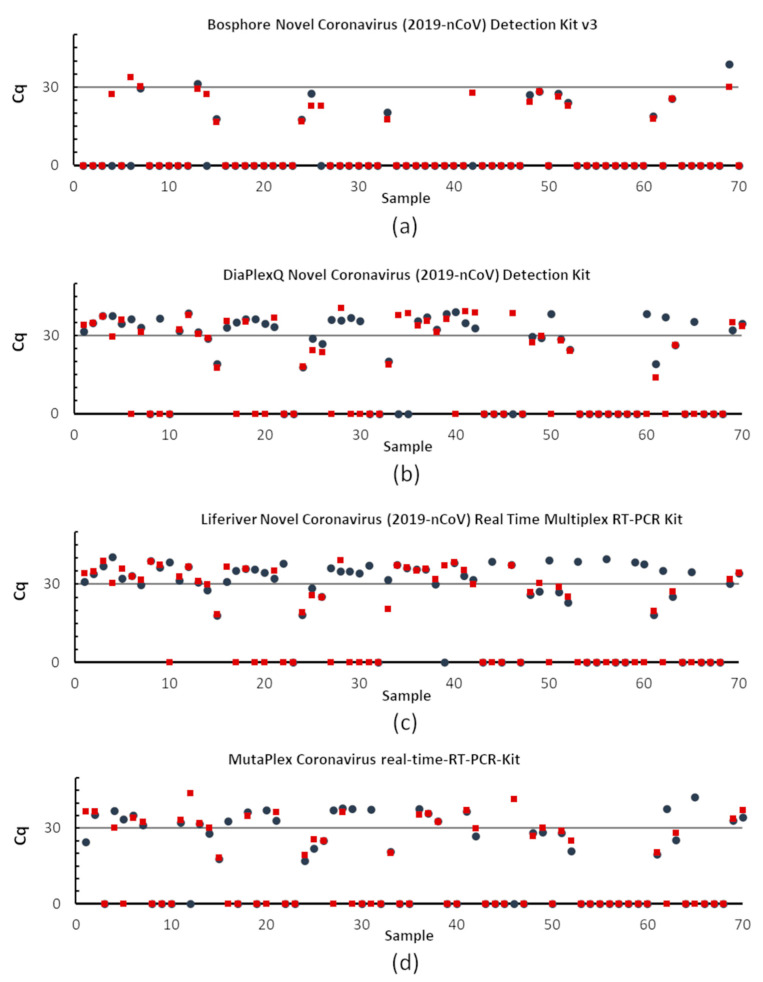
Diagnostic sensitivity impact of the applied RT-qPCR assays. RNA from the panel of 70 samples was extracted using column—(blue circles) and magnetic (red squares)-based methods. The graphs present Cq values for each specimen analyzed by four commercial RT-PCR kits detecting SARS-CoV-2, (**a**) Bosphore, (**b**) DiaPlexQ, (**c**) Liferiver, and (**d**) MutaPlex. Here, the Cq value is shown only for one of the tested regions, (**a**–**c**): Orf1ab, (**d**): RdRp—a fragment of Orf1ab gene; Cq equal to 0 means no amplification was observed.

**Table 1 diagnostics-11-02247-t001:** Manufacturer characteristics and results interpretation of SARS-CoV-2 RT-qPCR detection kits.

ASSAY	Targets	Results Interpretation		Controls
		POSITIVE	INCONCLUSIVE	
Bosphore Novel Coronavirus (2019-nCoV) Detection Kit v3	N-gene; Orf1ab; E-gene	Amplification of all (three) targets	Amplification of one or two targets	Internal Control
DiaPlexQ Novel Coronavirus (2019-nCoV) Detection Kit	N-gene; Orf1ab	Cq ≤ 40, at least one target amplified	Cq > 40	Internal Control
Liferiver Novel Coronavirus (2019-nCoV) Real-Time Multiplex RT-PCR Kit	N-gene; Orf1ab; E-gene	Cq ≤ 41, amplification of three or two targets or only of ORF1ab	Cq ≤ 41, amplification of only one target (excluding ORF1ab)	Internal Control
MutaPlex Coronavirus real-time-RT-PCR-Kit	RdRp/S-gene; E-gene	Amplification of one or two targets, wherein E-gene amplification also detects SARS-CoV-1 RNA	Not considered	Internal Positive Control; Sample Control

**Table 2 diagnostics-11-02247-t002:** RT-qPCR thermal profiles.

Kit	Desription	Time [min]	Temperature [°C]	Cycles
Bosphore	Reverse Transcription	17:00	50	1
	Initial Denaturation	06:00	95	1
	Denaturation	00:30	97	37
	Annealing/Extension	00:30	62	
DiaPlexQ	Reverse Transcription	15:00	50	1
	Initial Denaturation	15:00	95	1
	Denaturation	00:20	95	45
	Annealing/Extension	00:40	60	
Liferiver	Reverse Transcription	10:00	45	1
	Initial Denaturation	03:00	95	1
	Denaturation	00:15	95	45
	Annealing/Extension	00:30	58	
MutaPlex	Reverse Transcription	10:00	45	1
	Initial Denaturation	05:00	95	1
	Denaturation	00:10	95	45
	Annealing/Extension	00:40	60	

**Table 3 diagnostics-11-02247-t003:** A minimal number of detected SARS-CoV-2 copies.

	Applied RT-qPCR Kit			
No. of Copies/Sample	Bosphore	DiaPlexQ	LifeRiver	MutaPlex
1000	POSITIVE	POSITIVE	POSITIVE	POSITIVE
500	POSITIVE	POSITIVE	POSITIVE	POSITIVE
250	POSITIVE	POSITIVE	POSITIVE	POSITIVE
125	INCONCLUSIVE	POSITIVE	POSITIVE	POSITIVE
63	NEGATIVE	POSITIVE	POSITIVE	POSITIVE
32	NEGATIVE	INCONCLUSIVE	INCONCLUSIVE	INCONCLUSIVE
<16	NEGATIVE	INCONCLUSIVE	INCONCLUSIVE	NEGATIVE

**Table 4 diagnostics-11-02247-t004:** Reliability of the studied diagnostic tests. CI: confidence interval, PPV: positive predictive value, NPV: negative predictive value.

	Bosphore		Diaplex		Liferiver		MutaPlex	
	%	95% CI	%	95% CI	%	95% CI	%	95% CI
	column-based RNA extraction							
Sensitivity	34	21.5–48.3	96.2	87–96.2	98.1	89.9–99.9	94.3	84.3–98.8
Specificity	100	80.4–100	94.1	71.3–99.9	41.2	18.4–67.1	94.1	71.3–99.9
PPV	100	88.8–100	98.1	89.7–99.9	83.9	72.3–91.9	98	89.5–99.9
NPV	32.7	20.3–47.1	88.9	65.3–98.6	87.5	47.3–99.7	84.2	60.4–96.6
	magnetic-based RNA extraction							
Sensitivity	58.5	44.1–71.9	67.9	53.6–80.1	79.2	65.9–89.1	75.4	61.7–86.2
Specificity	100	80.5–100	94.1	71.3–99.9	94.1	71.3–99.9	94.1	71.3–99.9
PPV	100	88.9–100	97.3	85.8–99.9	97.7	87.7–99.9	97.6	87.1–99.9
NPV	43.6	27.8–60.4	48.5	30.8–66.4	59.2	38.8–77.6	55.2	35.7–73.5

**Table 5 diagnostics-11-02247-t005:** Analysis of statistical differences for the studied RT-qPCR tests expressed as a *p*-value for two RNA extraction methods, blue—column-based and red—magnetic-based. Comparisons for paired nominal data were performed using chi-square McNemar’s Marginal Homogeneity Test. A *p*-value < 0.05 is statistically significant.

	Bosphore	Diaplex	Liferiver	Mutaplex
Bosphore		0.07710	0.00150	0.00443
Diaplex	<0.0001		0.04123	0.22067
Liferiver	<0.0001	0.00938		0.61708
Mutaplex	<0.0001	1.00000	0.00257	

## Data Availability

All relevant data are within the manuscript.

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
