# Peer review of "Impact of the Nucleic Acid Extraction Method and the RT-qPCR Assay on SARS-CoV-2 Detection in Low-Viral Samples"

_diagnostics, 2021, doi:10.3390/diagnostics11122247_

Round 1

Reviewer 1 Report

I appreciate the efforts of the authors to review the content of the manuscript. It now appears significantly improved, clear and fluid in content. The manuscript should be ready for publishing. Congratulations.

Author Response

We thank the reviewer for comments and recommendations. 

Reviewer 2 Report

In this revised manuscript the Authors have provided valuable improvement in some sections (albeit amendment for typos is still required, e.g., Line 334) and in the rationale of the research. Unfortunately, major issues are still unresolved and the manuscript is still poorly prepared in key parts (e.g., materials and methods and resultl section). Some descriptive statistical measures have been provided, e.g., confidence interval (table 3) but there is no information on any formal analysis in support of the research hypotheses. A new section (“2.7 Statistics” L303) is now present, but is not providing clear information about statistical procedures. In L331-340 of the results, statistical differences are mentioned but it is not reported what type of test was conducted. Similarly, in L390-395, L455-461 and table 3  statistical comparisons are mentioned but eventually not provided.

Again, essential details on the RT-qPCR reactions are missing (for instance, primer concentration, temperatures and duration of the run, technical vs. biological replicates).

Author Response

We thank the reviewers for all their comments and recommendations. We rewrite our manuscript to meet the requirements of the reviewers.

Responses to the 2nd reviewer comments

Point 1: In this revised manuscript the Authors have provided valuable improvement in some sections (albeit amendment for typos is still required, e.g., Line 334) and in the rationale of the research.

Response 1: We screened the text again and corrected typos.

Point 2: Unfortunately, major issues are still unresolved and the manuscript is still poorly prepared in key parts (e.g., materials and methods and results section).

Response 2: Thanks for your comments. We analyzed the text again and tried to improve it according to your comments.

Point 3: Some descriptive statistical measures have been provided, e.g., confidence interval (table 3) but there is no information on any formal analysis in support of the research hypotheses. A new section (“2.7 Statistics” L303) is now present, but is not providing clear information about statistical procedures.
Response 3: The confidence intervals were calculated using the module Evaluation of a diagnostic test of PQStat software. According to the software developer, confidence intervals were calculated based on the Clopper-Pearson method for a single proportion. We add this information to the 2.8. Statistics section.

Point 4: In L331-340 of the results, statistical differences are mentioned but it is not reported what type of test was conducted.

Response 4: Thank you for finding this inaccuracy. We have already corrected the omission of the statistical description for the assay in lines 331-340. We added it to section 2.8. Statistics.

Point 5: Similarly, in L390-395, L455-461 and table 3 statistical comparisons are mentioned but eventually not provided.

Response 5: ‘The reliability of the studied diagnostic tests was analyzed using the module Evaluation of a diagnostic test, PQStat software, where confidence intervals were calculated based on the Clopper-Pearson method for a single proportion. As true positives, samples that were positive in at least two tests were considered. The total number of samples tested was 70, of which 52 were True Positives (TP) and 18 True Negatives (TN)’.

Point 6: Again, essential details on the RT-qPCR reactions are missing (for instance, primer concentration, temperatures and duration of the run, technical vs. biological replicates).

Response 6: We realize that the detailed description of the tests is very important. However, in our research, we use commercially available tests, for which the manufacturer does not provide values such as the concentration of primers and probes. We have the RT-qPCR thermal profilesfor each test in the form of Table 2. It was impossible to take the three samples from the patients, so we did not use biological replicates. This information was added to the described methods.
We believe that the methods we use are sufficient for diagnostics, which is the primary goal of this manuscript. However, the biological replicates could add to the scientific value of our manuscript, but we are not able to repeat experiments.

Round 2

Reviewer 2 Report

I have inspected this new version. Although I recognize the Authors' efforts in quickly amending the manuscript, major statistical flaws remain. 

For instance, the Authors answered "Response 3: The confidence intervals were calculated using the module Evaluation of a diagnostic test of PQStat software. According to the software developer, confidence intervals were calculated based on the Clopper-Pearson method for a single proportion.". This procedure is only for calculation of confidence intervals and not a statistical method for formal comprisons, which would be needed to test the research hypotheses that were stated in the introduction.   

In L139-143 is reported that "Data were obtained from three, two or single measurements, as defined in the specific sections, and were expressed as the arithmetic means ± standard deviation. Where it was possible, statistical analyses were performed using one-way ANOVA, followed by the Bonferroni-Holm post hoc test (19). Statistically significant differences between groups were defined as p-values less than 0.05." However results of this analysis were not reported in the result section (text, tables nor figures)

For this reason I strongly suggest the Authors to have a statistician providing ad-hoc mentoring for analyzing the interesting data that are presented in this manuscript. All this considering, I think that this manuscript cannot be considered for publication. 

Author Response

We thank the reviewers for all their comments and recommendations. We rewrite our manuscript to meet the requirements of the reviewers.

Responses to the reviewer comments

Point 1: For instance, the Authors answered "Response 3: The confidence intervals were calculated using the module Evaluation of a diagnostic test of PQStat software. According to the software developer, confidence intervals were calculated based on the Clopper-Pearson method for a single proportion.". This procedure is only for calculation of confidence intervals and not a statistical method for formal comprisons, which would be needed to test the research hypotheses that were stated in the introduction 

Response 1: Thank you for this remark. We fully agree with it. We have separated the part concerning test reliability from the statistics subsection.

Point 2: In L139-143 is reported that "Data were obtained from three, two or single measurements, as defined in the specific sections, and were expressed as the arithmetic means ± standard deviation. Where it was possible, statistical analyses were performed using one-way ANOVA, followed by the Bonferroni-Holm post hoc test (19). Statistically significant differences between groups were defined as p-values less than 0.05." However results of this analysis were not reported in the result section (text, tables nor figures)

Response 2: We have added the results in Figure 1

Point 3: For this reason I strongly suggest the Authors to have a statistician providing ad-hoc mentoring for analyzing the interesting data that are presented in this manuscript. All this considering, I think that this manuscript cannot be considered for publication.

Response 3: Thanks for your suggestion. We supported a statistician whom we added as a co-author of the publication. We performed additional statistical analyzes comparing the tests which were described in subsection 2.7 Statistical comparison of RNA extraction methods and RT-qPCR tests

Round 3

Reviewer 2 Report

The Authors addressed some changes. I still have the following comments:

Caption of figure 1: add information about the statistical test that supports the sentence "The red marker shows the sample for which the results differed between the RNA extraction method (red indicates positive, while the standard color is negative)."

Caption of figure 3: it is very synthetic, - please, include more information.

L252-253: all p-values must have the same number of digits.

Table 5: replace "0.00000" with "p < 0.0001".

Author Response

Point 1 Caption of figure 1: add information about the statistical test that supports the sentence "The red marker shows the sample for which the results differed between the RNA extraction method (red indicates positive, while the standard color is negative)."

Response 1: We have formulated this caption incorrectly, for which we apologize. The assumption of this description was not a statistical comparison of the obtained result. We have reworded this sentence to more accurately express our intentions. 

Point 2 Caption of figure 3: it is very synthetic, - please, include more information.

Response 2: We improved the description, but did not change it significantly as the additional information could be confusing. 

Point 3 L252-253: all p-values must have the same number of digits.

Response 3: Corrected

Point 4 Table 5: replace "0.00000" with "p < 0.0001"

Response 4: Corrected

This manuscript is a resubmission of an earlier submission. The following is a list of the peer review reports and author responses from that submission.

Round 1

Reviewer 1 Report

Komiazyk and colleaques present a comparative study on SARS-CoV-2 diagnostics via commercial qPCR tests with a comparison of RNA-extraction techniques. The study is very lean designed and focused.
The authors might want to correct some remaining typos e.g. in abstract (line 10, "SRAS-CoV-2") or improper sentence structure (line 61, "The used specimens were collected and analyzed anonymously.") 
Some major critical points are discernible:
The authors did not explain the choice of kits. They claim that the experiments were "performed using four commonly used commercial kits". What does this mean? The choice of kits (one Turkish, one Korean, one Chinese, one German manufacturer; all small companies) appears arbitrary. The results of Figure 1 are consequently only a quality control of these arbitrary picked tests and therefore of limited scientific value. 
The authors integrated a chapter on SARS-CoV-2 detection limits of the RT-qPCR assays, but failed to produce correct data, because they use incorrect control template that failed for most commercial kits. In remains unexplained, why the authors do not include a correct template that works for all testkits. In addition this chapter is written in an unclear and incomprehensible way and needs to be revised. This is continued in Table 2, where the majority of entries is not detected or no target, due to the insufficent RNA-template choice. The authors need to repeat the experiments with an correct template control and need to properly explain their choice.
The main results of this study is described in chapter 3.3. describing an possible impact of the RNA extraction method on the detection result. The results are hampered by the general layout of the study, which only includes two different ways of extraction methods. One column-based and one method based on magnetic beads. The author only measure RNA extracted with the two methods once with every qPCR kit. This narrow experimental design prevents a statistical evaluation of the experiment without which a valid statement can hardly be made. Multiple repetition of the experiments and basic statistical calculation (e.g. a simple mean value) can help out. In addition authors needs to add more methodical information or objectify if the experimental design is sufficient. They did not add any details on the RNA amount, RNA integrity or extraction control for input in the qPCR test. Was an extraction postive control used in both RNA extraction method? Was the amount of RNA output from the sample and input into PCR measured and normalized? This is mandatory and could easily be added to the manuscript to be valid and meaningful as scientific study.
Due to the layout of the study with sharp-shaped focus, the authors do not have a very comprehensive base for the discussion. They fill the discussion with general remarks on diagnostics on SARS-CoV-2 (e.g. lines 193-204 or 218-223) on possible improvement by implementing world-wide standards, doing repeats of positive tests or do proper reporting of results. All those remarks are well intentioned, but seem to mask the gaps of the layout of the study. 

Reviewer 2 Report

Impact of the Nucleic Acid Extraction Method and the RT- 2 qPCR Assay on the Detection of SARS-CoV-2 in the Diagnostic 3 Laboratory

Technical Comments to the Author

The manuscript is written in a clear and comprehensible manner, the results are more very fluent. Furthermore, the connection among the results and the figures/tables as well as final message is strong.

Remarks to the Author

I suggest a major and minor comments

Minor comments

  • I suggest reporting the purpose of the study in the introduction.
  • In line 81, replace 105 with 105
  • English language and style are fine/minor spell check required
  • Expand references in the study.

Reviewer 3 Report

Although I am not a specialist of SARS-CoV-2, in my opinion the manuscript needs subtantial improvement before it can be considered for publication in a high-quality journal. In particular, the relevance of the results in the exhisisting literature is not properly justified in the introduction and discussion (only 11 references were cited). Crucial protocol details are missing in materials and methods. The number of replications are sometimes missing. No formal analysis has been conducted.

L10: check the virus name

Introduction: It is not clear what would be the valuable addition of these results to the exhisting literature. There are hundreads of manuscripts dealing with SARS-CoV-2 detected using RT-qPCR and plenty of them focus on analytical/sensitivity aspects of diagnostic protocols

L32-41: In this long paragraph several information are reported but no literature references have been specified

L47: reference 8 is not Rezgar et al.,

Materials and methods: several important procedural details are missing. Also information about the adopted normalization procedure of RT-qPCR results are missing. Statistical analysis is missing.

L58: details on nasopharyngeal cavity swabs are missing

L67: correct the typo in “-80oC”

L67: should be “manifacturers’” because two brands were evaluated

L68-76: details about the RT-qPCR reaction (e.g., primer concentration, temperatures and duration of the run, reaction specificity) are missing

L83: How did you quantify the nucleic acids and estimated known number of gene copies?

L90: Are these technical or biological replicates? In any case, at least three biological replications are recommended for evaluating differences bewteen groups.

Results: Efficiency of RT-qPCR assays among commercial kits should be compared using formal analysis.

L111-112: Is written that “The MutaPlex and LifeRiver kits exhibited a similar limit of detection”. However, no formal analysis was conducted in support.

L118-144: This part is descriptive only – no statistical analysis has been conducted, hence the value of these results cannot be properly assessed.